# Experience and Awareness of Health Managers, Administrators, and Workers on a Hearing Conservation Program in Korea: A Qualitative Study

**DOI:** 10.3390/ijerph17072302

**Published:** 2020-03-29

**Authors:** Minsu Ock, Jeehee Pyo, Ok Hyun Kim, Changho Chae, Byeong Jin Ye, Sang Hoon Kim, Hyun Chan An, Ahra Kim, Jae Oh Park, Jiho Lee

**Affiliations:** 1Department of Preventive Medicine, Ulsan University Hospital, University of Ulsan College of Medicine, Ulsan 44033, Korea; ohohoms@naver.com (M.O.); eesther0517@naver.com (J.P.); 2Safety and Health Inc. Besh Consulting, Gimhae 50902, Korea; kalkihc@naver.com; 3Department of Occupational & Environmental Medicine, Sungkyunkwan University Samsung Changwon Hospital, Changwon 51353, Korea; chchae@naver.com; 4Department of Occupational and Environmental Medicine, Busan Paik Hospital & Institute of Environmental and Occupational Medicine, Inje University, Busan 47392, Korea; ong94@hanmail.net; 5Department of Occupational and Environmental Medicine, Ulsan University Hospital, University of Ulsan College of Medicine, Ulsan 44033, Korea; karlmanman@naver.com (S.H.K.); anhc008@naver.com (H.C.A.); 6Environmental Health Center, University of Ulsan College of Medicine, Ulsan 44033, Korea; ar0704@naver.com; 7Occupational Safety & Health Research Institute, KOSHA, Ulsan 44429, Korea; newnicx@kosha.or.kr

**Keywords:** workplace, qualitative awareness, hearing loss, hearing conservation program, focus group discussion

## Abstract

This study aims to evaluate the experience and awareness of a hearing conservation program to explore its activation plan. Three focus group discussions were conducted with five health managers, five labor supervisors, and five workers. A single in-depth interview was conducted with a health manager. Since hearing loss has a significant influence on the quality of life of workers, all participants recognized the importance of management. Although the need for hearing conservation programs was acknowledged, the participants had negative views about their effectiveness. Most health managers have not been able to demonstrate tangible results from their efforts to solve hearing problems, and they have been unable to demand that their employers actively invest resources and personnel in solving or preventing hearing problems. The participants in this study did not entirely understand the hearing conservation program, negative comments suggested that it is impossible to eliminate noise sources from the workplace, and measures for noise reduction would reduce work efficiency. This study can be supplied not only as a basis for reidentifying the real problems of the hearing conservation program but also for the tailored implementation method of future hearing conservation programs at each worksite.

## 1. Introduction

Hearing loss causes communication problems, which can lead to problems such as accidents, loss of productivity, and social isolation [1,2,3]. The prevention of hearing loss is significant, as recovery is nearly impossible, and it is necessary to cease the exacerbation if some hearing loss occurs. In the Republic of Korea (hereinafter Korea), about 9% of the total population is reported to have hearing loss on one side, and about 13% on both sides [4]. Therefore, hearing loss prevention is in the public’s interest.

Workplace noise is one of the most common causes of hearing loss [5,6]. In Korea, there was a total of 4774 excessive noise workplaces nationwide in 2015, with a total of 10,125 persons with noise-induced hearing loss, which averages to about 2.12 persons with a noise-induced hearing loss per workplace [7]. The number of people with noise-induced hearing loss according to regional distribution was highest in Gyeongsangnam province (2738 cases), followed by Ulsan metropolitan city, Gyeonggi province, and Busan metropolitan city. Ulsan had the highest number of persons with a noise-induced hearing loss per workplace, with 3.29 persons on average. Gangwon province had 3.05 persons per workplace, and Gyeongsangnam province had 2.33 persons per workplace. These statistics illustrate the seriousness of manufacturing workplace-related hearing loss in several regions in Korea.

A hearing conservation program is stipulated in rule 517 of the Occupational Safety and Health Standard to prevent hearing loss of workers in Korea [8]. The hearing conservation program is a comprehensive plan for the prevention and management of noise-induced hearing loss due to noise generated in the workplace. It includes noise exposure surveys, engineering control for noise sources, management of hearing protection devices, preventive education and motivation, regular hearing screening and follow-up, and documentation and management [9]. Rule 517 states that establishments with noise levels exceeding 90 decibels according to the workplace noise measurement and those with cases of noise-induced hearing loss in the workplace are required to establish and implement a hearing conservation program. The Korea Occupational Safety and Health Agency has also developed and distributed guidelines for instituting, implementing, and evaluating the effectiveness of hearing conservation programs in the workplace [9].

Despite the regulations and related guidelines for workers exposed to noise, however, the number of diagnosed cases of occupational noise-induced hearing loss from special health examinations has increased every year in the last four years: 6684 cases in 2012, 7388 in 2013, 8428 in 2014, and 10,042 in 2015 [7]. The most frequent occurrence of noise-induced hearing loss (D1) according to the scale of the company was seen in small-scale establishments consisting of 5–29 workers, followed by establishments with 299 or fewer workers. As categorized by the type of industry, the manufacturing industry had the most diagnosed cases with 8202 people (accounting for 81% of the total 10,125 workers), then, construction with 1006 workers, mining with 183 workers, repairs with 170 workers, and transportation with 164 workers.

The implementation of a hearing conservation-related program in the workplace is an obligation of employers and can prevent and reduce the incidence of noise-induced hearing loss [10,11,12]. Current legislation of hearing conservation is, however, theoretical rather than practical, and a lack of individual working environment considerations prevents compliance with the hearing conservation program [13]. There is an expectancy of a constant outbreak of people with noise-induced hearing loss, as the existing regulations and guidelines are not adequately applied in the field. If the situation persists, even if a variety of hearing conservation programs are offered in the future, the difficulty of engaging the establishments will continue to increase the number of persons with hearing loss.

This study aims to evaluate the experience and awareness of the hearing conservation program to explore its activation plan. Specifically, we gathered opinions on experiences, problems, and facilitation methods of hearing conservation programs in the workplace through focus group discussions with workers, health managers, and labor managers to determine ways to improve hearing conservation programs.

## 2. Materials and Methods

Qualitative research was executed using focus group discussions and in-depth interviews to evaluate the experiences and perceptions of hearing conservation programs. We used the consolidated criteria for reporting qualitative research (COREQ) using an in-depth interview or focus group discussion, as described below [14].

### 2.1. Ethics Approval and Consent to Participate 

This study was approved by the Institutional Review Board of Ulsan University Hospital (Project Number: 2018-07-004). All participants were informed about the purpose and process of the study and only those who agreed to participate joined this qualitative study.

### 2.2. Research Team Composition

The research team consisted of nine persons. Two of the researchers have extensive experience in qualitative research, such as writing several theses and articles using the methodology, regularly attending qualitative research-related lectures and seminars, and teaching lectures. Seven of the researchers are either occupational and environmental specialists or have pervasive experience in occupational and environmental-related work and research, especially in noise problems. Four researchers are professors, three are residents of occupational and environmental medicine, and three are institution researchers.

### 2.3. Research Participants

Purposive sampling was used to recruit research participants, no unique inclusion and exclusion criteria were set up, and participants were invited to provide detailed expert opinions and practical experiences on research topics. Five health managers, five labor supervisors, and five workers participated in the focus group discussion. A single in-depth interview was conducted with a health manager. The participant who was a health manager in a company with a relatively decent performance status of the hearing conservation program was selected for the in-depth interview. Separate interviews were conducted with participants who had successful experiences in the hearing conservation program. This interview design considered the possible influence of the successful experience of the program to the other participants in the focus group discussion.

Researchers individually contacted and explained the purpose and content of the study to selected health managers and workers who agreed to participate in the study’s focus group discussions. The labor managers agreed to participate in the study with the cooperation of the Ministry of Employment and Labor and the Korea Occupational Safety and Health Agency. Consent was obtained from all research participants. No participant refused to participate in the study, nor did any drop out. The reflexivity, the possibility of responding to researchers’ preference, was evaluated to appear in the perspective of the research topic and purpose unlikely.

### 2.4. Focus Group Discussions and In-Depth Interview

Discussions and interviews were conducted using semi-structured guidelines developed through researcher discussions, reviewing primary precedent studies, and hearing conservation program descriptions [9]. The guidelines included awareness of hearing loss problems, experience, and overall perception of hearing conservation programs, the opinions about each component of the hearing conservation program, and more are shown in the Appendix A. The content of opinions of the hearing conservation program components was, however, adjusted to the characteristics of the study participants.

All of the focus group discussions and interviews were conducted in separately rented meeting rooms. Written informed consent was obtained from the participants after a verbal explanation of the study was given before the discussions and the interview proceeded. Two researchers executed discussions and interviews. One researcher has numerous experience in a focus group discussion facilitation, and the other has extensive research experience in hearing loss. The discussions and interviews took about two hours each. We attempted to elicit genuine opinions of the participants through sufficient ice-breaking time before the official focus group discussion and interviews. Additionally, the researchers tried to hide their opinions during data collection. They reviewed their prejudice or bias of the research before the execution of the focus group discussion and interviews. The discussions and interviews were recorded for transcription. A stenographer transcribed the content of focus group discussions and interviews, and the facilitator reviewed the content for any missing material. Additionally, the facilitator recorded a filed note during the process of the focus group discussions and interviews. The note was used to clarify the meaning of the participants’ statements.

### 2.5. Analysis

Content analysis of qualitative research was used for analysis [15]. The analysis utilized the transcribed contents of the discussions and the memorandum written by the researcher during the discussion and interview process. Content analysis is a method of extracting, reinterpreting, and deducing crucial implicit meanings embedded in original data, by using pre-set theories or perspectives. In this study, direct content analysis was performed, which refers to the deductive categories derived from existing theories or research results in the analysis [15]. No separate software was used for the analysis, and Microsoft Word was utilized for the transcription.

Using the accurate analysis method, one researcher repeatedly read the transcribed contents and memo and derived concepts, and then one of the other researchers reviewed the concepts. Afterward, the two researchers who participated in the primary analysis categorized similar concepts among the ones derived by agreement. All of the researchers then checked the results of the analysis, focusing on the framework of the whole category. The two researchers concluded the analysis after confirming the data saturation, which does not create any further concept. The research participants did not review the transcriptions and results of the data analysis, but experts in qualitative research, occupational and environmental medicine, and preventive medicine reviewed it.

## 3. Results

### 3.1. Socio-Demographic Characteristics

Five out of the six health managers participated in the focus group discussion, and one participated in the in-depth interview. The details of the socio-demographic characteristics of participants are shown in Table 1.

### 3.2. Analysis Result

A total of 390 codes were derived from the analysis and categorized according to the awareness of the hearing loss problem, the overall perception of hearing conservation programs, the opinions of each component of the hearing conservation program, and the plan to vitalize the hearing conservation program. Table 2 summarizes the results of the analysis; essential contents are described according to each category and subcategory.

#### 3.2.1. The Awareness of Hearing Loss Problems

1) Lack of voluntary interest in hearing loss problems

Overall, the participants acknowledged that they lacked interest in hearing loss. The health managers felt that workers were not interested in hearing problems, and the labor supervisors claimed that they focused on death and injuries rather than hearing problems. However, workers were more concerned about hearing problems than before.

I know there is a problem because we tend to focus more on death, accident, and injury, but it is quite demanding for us just to invest all our capacity to solve the noise-induced deafness issue. To be honest, the noise-induced hearing loss is not death by post-industrial accident. (Labor Supervisor 5)

2) Importance of hearing loss problem

Almost all the participants were aware that hearing loss was a significant health management issue. In particular, most health managers, labor supervisors, and workers have acknowledged that hearing loss is a crucial problem as it significantly deteriorates the quality of life of workers.

I guess everything else too, but hearing is really related to the quality of life. There is a risk to any other disaster, but besides that, let us assume that I have been working in the heavy industry for 30 years (with hearing loss). And after retirement, my wife and I sit on the couch, but we can’t watch TV together. How low is the quality of life you have? Your family doesn’t have a conversation with you, with dad because he can’t hear you and understand you. When children watch a TV with dad, they tell him to watch it somewhere else because they can’t turn up the volume. So this guy is always lonely. He becomes a loner. It is not only just hearing loss, but also the quality of life itself gets totally worse. It does not compensate for any disability rating and rewards.(Health Manager 2)

3) Difficulty in solving hearing loss problems

The health managers were, directly and indirectly, experiencing the problem that hearing loss was not easily exposed and difficult to improve with engineering solutions. Of this reason, the labor supervisors were not able to actively manage and supervise the delicacy of solving the hearing loss problem, and it was complicated for the workers to clarify the issues of hearing loss as they were aware of the trouble of enhancing the working environment in reality.

It’s too expensive to upgrade it. I could only educate them (workers), give them some earplugs, and do some hearing tests.(Health Manager 4)

No one really cares about it, either they are big or small companies because it’s not on the surface level.(Worker 2)

#### 3.2.2. The Overall Perception of the Hearing Conservation Program

1) Priority issues of hearing management

From the health managers’ perspectives, hearing management problems were not a priority as the high volume of other tasks besides noise-related ones. On the other hand, the workers thought that hearing management was not at the top of the list as there are no immediate symptoms for hearing loss, and they have hearing aids for hearing deterioration alternatives.

If you hurt your eyes, then you go blind and can’t see right away, so it becomes a problem. But with ears, wearing hearing aids solves the problem on the spot, so it’s not on the top of the list. Nobody talks about it, so nobody makes a problem of it.(Worker 2)

2) Importance of workers’ hearing problems to health managers

It was substantial to the health mangers to prevent the occurrence of D1 workers with difficulty in hearing. If any incidence of D1 is verified, the level of supervision and supervision of the Ministry of Labor can be strengthened. Therefore, the employer and the health managers were concerned with preventing the D1 emergence.

If the D1 comes out, the company is interested in it, but not with C1 because they think that it’s not that big of the deal or they think it could happen.(Health Manager 4)

3) Hearing conservation program not yet feasible

The health managers felt that hearing conservation programs were inefficient as they had not witnessed evident cost-effectiveness from the previous investment. This experience discouraged them from demanding further resources from their employers actively. The labor supervisors, however, were worried that hearing loss problems were not well expressed in workplaces where hearing loss problems were serious, and hearing conservation programs were not applied well in paradoxically serious workplaces.

Noise is our top priority because it is one of the themes of the work environment improvement. The noise is too loud, so there is a section for noise in the task objectives, and we also have a separate theme for it. There is no direct cost-effectiveness for the noise part, so it’s not so easy to touch upon that. And when you actually work on it, just simply saying "improve it!" won’t do much, so the noise section is the trickiest one.(Health Manager 4)

#### 3.2.3. The Opinions of Each Component of The Hearing Conservation Program

Hearing conservation programs were broadly divided into noise measurement, engineering control of noise, wearing the protective device, measuring of hearing threshold, health education, and documentation and effect evaluation.

1) Noise measurement

The health managers were receptive to the importance of noise measurement. They were attempting to measure noise by themselves or outsourcing, and endeavoring to demonstrate the results in the form of noise signs visually. The noise was not measured; however, every time the process was changed, the noise mapping was hardly achieved.

I put a noise sign up at a noisy facility, and I wrote down all the figures so that everyone is aware that ’Noise in this area is 90.1 decibels.’ I put it all over the noisy place. When I first put it up, some workers told me nothing good would come if the Ministry of Labor sees the over 90-decibel numbers visibly. Now, when the workers see it from a distance, they think ‘That field is over 100 dB. I should wear earplugs because it would be loud there.’ I did it anyway so that at least they could be aware of it and wear a protection device.(Health Manager 4)

2) Engineering control of noise

The health managers recognized that engineering control is significant and pertained experience of solving noise problems at a low cost. Of cost problems, however, it was acknowledged that the engineering control could not solve all noise problems. Although the labor supervisors believed that engineering control is more economical from a broad perspective, they suggested that it would be difficult to share the engineering control between workplaces regarding intellectual property rights. Workers, nonetheless, speculated that engineering control of noise is impossible and had not expected the engineering control of noise via the experience of work efficacy diminution due to the noise reduction measure.

As you said, it becomes an intellectual property for the establishment when they invest lots of money and get consulted to take measures. I think it’s a bit too much to force them to share that... (Labor Supervisor 3)

We use a urethane hammer, and it cost far more than a sledgehammer because when you pound the urethane one a few times, it cracks. Besides the cost part, the urethane hammer does not work for us realistically. You have to give it a big hit. When you hit a sledgehammer, you hit it once, but with the urethane hammer... It means you have to pound 10, 20 times.(Worker 3)

3) Wearing a protective device

The health managers expressed their frustrations that regardless of their strong recommendation to wear a protective device, such as earplugs, to prevent hearing loss problems, workers do not adequately wear even the protective device. There were some incidents where appropriate protective device wearing occurred via fitting tests. The workers wanted somewhat competent and individually suitable earplugs without inconvenience for wearing.

Earplugs, I mean, people all have different shapes of ears. You put the earplugs deep inside of your ears. You put it in, hold it for 1 or 2 seconds, and let it go when they inflate. Some people can’t wear it for long if they put it in like that. So some people can’t put it all the way in, just around the hole. Cause it’s painful.(Worker 2)

4) Measuring of hearing threshold

The health managers did not perceive repetitive hearing screening as a fundamental measure to solve the hearing problem, and it was challenging to deal with workers who thought severely of hearing screenings that did not offer any countermeasures. The workers also had the unfortunate experience of receiving proper guidance and measures after hearing tests.

The workers didn’t like it. They said they already know their ears are wrong, so why should they continue to do the same test. They complained that they would rather have a practical solution.(Health Manager 1)

(So you have not received any special instructions about ear screening. And how much should you work according to your hearing?) No, I have not.(Worker 1)

5) Health education

The health managers stated that the training through the fitting test was effective, and the managers thought that training for the employer as well as the workers was necessary. The workers said, however, that the actual training was a rough reference to the importance of wearing earplugs and they did not receive any training in how to wear the earplugs. Therefore, they either read the manual or learned from their colleagues.

The most motivating part of workers during the fitting tests was that they got to experience the effect of adequately wearing the device. We taught them that when they wear earplugs the way they are used to, the insulation of sound was only 5 to 10 dB. But they saw that the proper way of wearing the earplugs could create about 33 dB insulation of sound. We showed them that…(Health Manager 2)

There was no education, and nothing comes out when you actually get it from the company. They just say wear earplugs properly. That’s all.(Worker 4)

6) Documentation and effects evaluation

Most participants were not engaged in documentation and evaluating effects. Although some health managers supervised the hearing levels of workers individually, most of them relied on documentation from screening institutions, and workers relied on memories without recording test scores. Likewise, the effectiveness of hearing conservation programs has not been appropriately evaluated.

(Do you know how much of your hearing level changed in the past?) I kind of do. I get the figures twice a year. Whenever I receive it, I think to myself ‘last year I was 45 dB, and this year it is 50 dB. It got a little worse.’ That’s how you think. I do not have separate documentation for the figures.(Worker 2)

#### 3.2.4. The Plan to Vitalize the Hearing Conservation Program

1) Willingness and attitude toward health manager’s hearing problem solving

Nonetheless, it was suggested that health managers should have the willingness to solve the noise problem and that they should demonstrate proactive self-studying and endeavoring attitudes. There were also expectations that the health managers would be able to resolve the problem of hearing management more efficiently if they depend on the authority of law.

They have to study; they need to study. … At least for the short-term thing, go to an educational center. No falling asleep, get all the questions you had, and just chat with professors. Say “I don’t know nothing, what do I do?” just go and ask and ask again to the experts who measure… "What do I do? I got no idea…"(Health Manager 6)

2) The necessity of management for subcontractors and small scale enterprises

The labor supervisors acknowledged that the problems of subcontractors and small scale enterprises were more critical and noted that financial support and management for small scale enterprises are essential. The workers also thought that large companies should pay attention to the environment of subcontractors and small scale enterprises.

The way you can enforce the domineering measure on small scale enterprises is that you first have to either assist them with the clean-business program or give them a monetary support measure. You can’t really cover all the workers working with noise in every establishment with prevention to follow up care. It’s not going to be like categorizing businesses by the noise level and give a different tailored solution. That is the biggest problem for me.(Health Manager 1)

3) Willingness to solve hearing problems of employers

The health managers and the labor supervisors considered the awareness and willingness of the employer were the most significant to solve the hearing problem.

I think all business owners must have a willingness to change and to move everything forward. The hearing conservation program also requires the willingness of the top executive to actualize it.(Health Manager 2)

4) The necessity of strengthening the administrative and legal system

There were several comments from the work managers and workers that it was necessary to legally enforce and regulate further hearing conservation programs for its poor performance. Additionally, it was suggested that a health manager should be a full-time employee for them to have enthusiasm and willingness, in genera, l to work on things such as noise problems.

The reason the program doesn’t work is related to the hiring condition of the health manager. Managers who need to quit their job every two years can never make it work. Health manager? There are numerous temporary workers. It doesn’t let you do any work. I could never have a devotion to my company, and it’s not even about the work environment. It’s impossible to go to the field when you can’t even do the simple screening and cover all the incoming patients. You don’t get to visit the field until you quit your job.(Health Manager 6)

It means that if engineering control is not taken, it will be repeated, and even if the program is operated, it will be a program with the same information even after several years. So in that part, if the actual employers do not open their wallet, the countermeasures cannot be made, and I think that those are the parts where you need to regulate the law a little harder.(Labor Supervisor 4)

5) Improving the hearing conservation program for a smooth application

The health managers claimed that easy to understand and use of the hearing conservation program is needed. The labor supervisors, however, believed that in order to revitalize the hearing conservation program, it is imminent to focus on advancing the practicality of the hearing conservation program rather than improving the contents itself. Moreover, the workers suggested the need for the program to consider the difficulties in the field.

I wish there are realistic ways to easily access the programs or simplify or replace the procedures for health managers or labor supervisors in the field.(Health Manager 4)

Basically, no matter how well the hearing conservation program is made, it’s meaningless if the worker or employer in the field has a hard time to apply it or do a quick operation for it.(Labor Supervisor 4)

## 4. Discussion

This research is significant because the perception and experience of hearing conservation programs were examined from various points of view, including health managers, labor supervisors, and workers. In addition, national studies on hearing conservation programs are scarce in quantity, and we know of no studies confirming the views of the stakeholders on hearing conservation programs. A qualitative research methodology was utilized for the first time in Korea to hear not only the overall awareness of hearing loss problems and hearing conservation programs but also the specific components of hearing conservation programs. As the qualitative research methodologies are known for identifying problems that have not yet been elucidated and understanding experiences or perceptions that are not well represented by numbers [16], this study’s research methodology is valid.

The principal finding of this study was a lack of interest in hearing loss problems, and that the problem of hearing loss is not a top priority for health managers at the workplace. However, since hearing loss has a significant influence on the quality of life of workers, the health managers, labor supervisors, and workers all recognized the importance of management implementing a hearing conservation program. Therefore, for the invigoration of hearing conservation programs, it is necessary to convey that hearing loss problems are a priority in workplace health management and inform the industry of the immense problem [17]. It is imperative to emphasize the fact that the number of workers who are diagnosed with occupational noise-induced hearing loss increases annually and to highlight the decline in quality of life due to hearing loss [1,2] and the decrease in work efficiency [3].

Although the need for hearing conservation programs was generally acknowledged, there was a prominent negative perspective about their effectiveness. Most health managers have not been able to demonstrate tangible results from their efforts to solve hearing problems, and they have been unable to demand that their employers actively invest resources and personnel in solving or preventing hearing problems. However, one of the participants in this study who is performing well in the hearing conservation program showed that hearing loss problems could be minimized if the hearing conservation program is adequately performed with constant attention expressed. It is essential to actively share and promote the experience of a workplace that has successfully implemented a hearing conservation program to other workplaces [11,18,19]. As one labor supervisor pointed out, sharing engineering control between workplaces can be strenuous in reality regarding intellectual property rights. Therefore, it is necessary for the central institutions, such as the Occupational Safety and Health Administration, to collect the outstanding engineering control with appropriate compensation for the workplace, and actively share and promote these measures in other workplaces.

The participants in this study did not understand the whole contents of the hearing conservation program. They were, however, aware of partial elements of the hearing conservation program, such as engineering control of the noise source, wearing hearing protection devices, fitness tests, and more. The result of the programs, the measurement of hearing thresholds, and their use in the audiometric database seem to be lacking. In particular, health managers need to re-measure noise when the process changes, but such mandatory procedures achieved insufficient performance. Moreover, the negative comments suggested that it is impossible to eliminate the noise source even in the case of workers, and measures for noise reduction would reduce work efficiency. Furthermore, most research participants were not involved in documentation and effect evaluation. It is necessary to perform proper training on each component of the hearing conservation program to benefit from the effects of the program [13].

For practical application of the hearing conservation program, however, it is imminent to propose “application of a customized program at the workplace” by creating a program suitable for the size and the specific reality of the workplace rather than a theoretically complicated process. As many health managers have indicated, it will be necessary to develop and distribute a hearing conservation program that is easy to understand and activate. Therefore, it is necessary to develop and present examples of hearing conservation programs that are easy to apply and specific to the workplace.

In addition to the improvement of the hearing conservation program itself, there was an emphasis on strengthening the willingness of the health manager and the administrative and legal system of the workplace. Specifically, it was suggested that employing a health manager as a full-time employee would enable him or her to manage the overall tasks of health care, including noise issues, with enthusiasm and passion. Moreover, it is necessary to consider offering incentives to the workplaces that grant authority to health managers. Some labor supervisors and workers mentioned that if the hearing conservation program is not appropriately performed despite these measures, the government will have to enforce and regulate the implementation of hearing conservation programs. Furthermore, as in previous studies highlighting the health inequalities in hearing loss [20], financial support and management of subcontractors and small businesses are required for its potential.

One of the limitations of this study is that it only includes one health manager who has been successful in implementing a hearing conservation program. In future studies, it is necessary to include more people with successful experiences. The sharing and dissemination of successful hearing conservation programs will increase the awareness of effectiveness as well as the need for hearing conservation programs. Another limitation is that the entire transcriptions and the analysis content were not reviewed by the participants. Instead, qualitative research experts, occupational and environmental medicine experts, and preventative medicine experts reviewed the content to increase its validity.

## 5. Conclusions

In this study, focus group discussions and an in-depth interview were conducted to review the experiences and perceptions of health managers, labor supervisors, and workers on hearing conservation programs. We found that the overall level of awareness of stakeholders on hearing conservation programs is not complete. In addition, this research shows that improvement of health managers’ attitudes, management of subcontractors and small business sites, employers’ willingness, and administrative and legal systems are important for the improvement of hearing conservation programs. This study can be supplied not only as a basis for reidentifying the real problems of the hearing conservation program but also for the tailored implementation method of future hearing conservation programs at each worksite.

## Figures and Tables

**Table 1 ijerph-17-02302-t001:** Socio-demographic characteristics of participants.

Classification	No.	Sex	Age Group	Years of Experience	Participation Category
Health manager	1	Female	50–59	32	Focus group discussions
	2	Female	40–49	27	Focus group discussions
	3	Female	40–49	20	Focus group discussions
	4	Female	20–29	7	Focus group discussions
	5	Female	30–39	17	Focus group discussions
	6	Female	40–49	23	In-depth interview
Labor supervisor	1	Female	40–49	7	Focus group discussions
	2	Male	40–49	7	Focus group discussions
	3	Male	30–39	4	Focus group discussions
	4	Male	40–49	11	Focus group discussions
	5	Male	40–49	7	Focus group discussions
Worker	1	Male	40–49	25	Focus group discussions
	2	Male	40–49	17	Focus group discussions
	3	Male	40–49	25	Focus group discussions
	4	Male	40–49	19	Focus group discussions
	5	Male	30–39	14	Focus group discussions

**Table 2 ijerph-17-02302-t002:** Structure of the analysis results and the main content.

Category	Subcategory
1. The awareness of hearing loss problems	1-1. Lack of interest in hearing loss problems
	1-2. Importance of hearing loss problems
	1-3. Difficulty in solving hearing loss problems
2. Overall perception of the hearing conservation program	2-1. Priority issues of hearing-care
	2-2. Importance of worker’s hearing problem to health manager
	2-3. Hearing conservation program not yet feasible
3.The opinions on each component of the hearing conservation program	3-1. Noise measurement
	3-2. Engineering control of noise
	3-3. Wearing hearing protective devices
	3-4. Measuring of hearing threshold
	3-5. Health education
	3-6. Documentation and effects evaluation
4.The plan to vitalize the hearing conservation program	4-1. Willingness and attitude toward health manager’s problem solving
	4-2. The necessity of management for subcontractors and small scale enterprise
	4-3. Willingness to solve hearing problems of employers
	4-4. The necessity of strengthening the administrative and legal systems
	4-5. Improving the hearing conservation program for a smooth application

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
