# Peer review of "Experience and Awareness of Health Managers, Administrators, and Workers on a Hearing Conservation Program in Korea: A Qualitative Study"

_ijerph, 2020, doi:10.3390/ijerph17072302_

Round 1
Reviewer 1 Report
Your objective(s) are not clear. It is suggested that you make use of the language editor.
The research design needs more information. it is not clear how your participants were recruited, the inclusion and exclusion criteria were not mentioned. Ethical considerations were not mentioned. furthermore, there is a concern that some participants were coerced to participate based on the approval of the Ministry of Labour. The rigour of the study was not discussed. Data analysis is not clear. it needs to be rephrased.
Since the objectives of the study were not clear, it was hard to follow the results, hence I could not proceed with the review. Please consider making use of the language editor's services.
Author Response
Comments and Suggestions for Authors
Your objective(s) are not clear. It is suggested that you make use of the language editor.
Response: We would like to thank you for reading our manuscript and reviewing it. The followings are point-by-point responses to your comments. As described in the Introduction, the purpose of this study was to evaluate the experience and awareness of hearing conservation program to explore the activation plan of hearing conservation program (Line 100~). We carefully read the manuscript again and revised the manuscript so that the purpose of the study can be clearly revealed. Although we have already received English editing, but have received an additional English editing.
Also, to clarify the our conclusions we change the sentence into “This study can be supplied not only as a basis for reidentifying the real problems of the hearing conservation program, but also for the tailored implementation method of future hearing conservation programs at each worksite.”
The research design needs more information. It is not clear how your participants were recruited, the inclusion and exclusion criteria were not mentioned. Ethical considerations were not mentioned. Furthermore, there is a concern that some participants were coerced to participate based on the approval of the Ministry of Labour. The rigour of the study was not discussed. Data analysis is not clear. it needs to be rephrased.
Response: Based on the comments of other reviewers, we have revised the method section to reveal the research methodology in more detail. In specific, more descriptions have been added on how to select research participants and their inclusion and exception criteria (Line 122~). Also, as you mentioned, it cannot exclude the possibility of involuntary participation of labor managers as we have asked for the cooperation of “the Ministry of Employment and Labor and the Korea Occupational Safety and Health Agency” for the participant recruitment. We had, however, exerted our utmost effort to respect the participants’ own volition in the process of the recruitment and only involved people who had consented to research.
Since the objectives of the study were not clear, it was hard to follow the results, hence I could not proceed with the review. Please consider making use of the language editor's services.
Response: As stated in the introduction, the purpose of this research is to evaluate the experience and awareness of the hearing conservation program to explore the activation plan of the hearing conservation program. We have reviewed and made edits on the entire manuscript and received additional English edit.

Reviewer 2 Report
This manuscript from the research group led by Dr. Jiho Lee evaluated the experience and awareness of the hearing conservation program and explored the activation plan of hearing conservation program. The authors demonstrated the novelty of this manuscript is that can be applied not only as a basis for revising and improving the hearing conservation program but also for the quantitative recognition of future hearing conservation programs. The conclusion is supported by the interviewed and analyzed results. The health managers, labor supervisors, and workers were in-depth interviews in this manuscript are logically described. However, only 15 person’s interviews were analyzed in this study, more research participants were needed to address the scientific conclusion.
Author Response
Comments and Suggestions for Authors
This manuscript from the research group led by Dr. Jiho Lee evaluated the experience and awareness of the hearing conservation program and explored the activation plan of hearing conservation program. The authors demonstrated the novelty of this manuscript is that can be applied not only as a basis for revising and improving the hearing conservation program but also for the quantitative recognition of future hearing conservation programs. The conclusion is supported by the interviewed and analyzed results. The health managers, labor supervisors, and workers were in-depth interviews in this manuscript are logically described. However, only 15 person’s interviews were analyzed in this study, more research participants were needed to address the scientific conclusion.
Response: We would like to thank you for reading our manuscript and reviewing it. The followings are point-by-point responses to your comments. This research investigated the experience and perception of the hearing conservation program via qualitative research methodology. This methodology strives to present in-depth narratives and stories of the lesser number of participants. Also, the methodology studies participants from one person to more than dozens as there is no definite rule on the sample size of the methodology—qualitative research, like ours, endeavors to explore the essence of the stories of participants. On the contrary, quantitative research attempts to generalize a phenomenon through a study with the high-volume of participants. We would like to ask you to consider the differences between the research methodologies. Also, to clarify the our conclusions we change the sentence into “This study can be supplied not only as a basis for reidentifying the real problems of the hearing conservation program, but also for the tailored implementation method of future hearing conservation programs at each worksite.”

Reviewer 3 Report
A. The quality of the English is poor enough as to make the reading difficult. It requires effort to understand the ideas the authors are trying to convey.
B. The organization of the paper is poor. Analysis are plagued with statements in a way that are not acceptable for a paper
C. The text is too extensive for its content. It can easily be halved.
D. The conclusions cannot be generalized from the sample size used. They are OK for the particular cases being analyzed.
In summary, the paper has to be revised using the help of persons familiar with writing scientific papers and with the help of somebody familiar with English language.
Author Response
Comments and Suggestions for Authors
- The quality of the English is poor enough as to make the reading difficult. It requires effort to understand the ideas the authors are trying to convey.
- The organization of the paper is poor. Analysis are plagued with statements in a way that are not acceptable for a paper.
- The text is too extensive for its content. It can easily be halved.
- The conclusions cannot be generalized from the sample size used. They are OK for the particular cases being analyzed.
In summary, the paper has to be revised using the help of persons familiar with writing scientific papers and with the help of somebody familiar with English language.
Response: We would like to thank you for reading our manuscript and reviewing it. The followings are point-by-point responses to your comments. We have reviewed and edited the entire manuscript and received additional English edits based on your suggestion. We would like to ask you to consider that this research used qualitative research methodology. This methodology strives to present in-depth narratives and stories of the lesser number of participants. Also, the methodology studies participants from one person to more than dozens as there is no definite rule on the sample size of the methodology—qualitative research, like ours, endeavors to explore the essence of the stories of participants. On the contrary, quantitative research attempts to generalize a phenomenon through a study with the high-volume of participants. We would like to ask you to consider the differences between the research methodologies.

Reviewer 4 Report
Hola,
Creo que el artículo debería publicarse si se realizan revisiones menores.
Los autores del estudio afirman haber establecido los criterios de calidad de acuerdo con las recomendaciones de COREQ, pero algunos de ellos no se han cumplido, como la revisión de los participantes y otros no se han explicado claramente. Se requiere una explicación adicional de cada uno de los siguientes criterios COREQ:
- Entrevistador / facilitador: ¿Qué autor / es realizó la entrevista o grupo focal?
- Ocupación: ¿Cuál era su ocupación en el momento del estudio?
- Género: ¿El investigador era hombre o mujer?
- Relación establecida: ¿Se estableció una relación antes del comienzo del estudio?
- Características del entrevistador: ¿Qué características se informaron sobre el entrevistador / facilitador? Por ejemplo , sesgos, suposiciones, razones e intereses en el tema de investigación.
- Methodological orientation and Theory: What methodological orientation was stated to underpin the study? e.g. grounded theory, discourse analysis, ethnography, phenomenology, content analysis
- Setting of data collection: Where was the data collected? e.g. home, clinic, workplace
- Description of simple: What are the important characteristics of the sample? e.g. demographic data, date. The researchers only provide information about the sex and age of the participants. It is recommended to provide information on the years of work experience of the participants regarding the phenomenon of study in order to favor the transferability of the data.
- Interview guide: Were questions, prompts, guides provided by the authors? Was it pilot tested? A table or annex with the semi-structured interview questions is recommended.
- Audio/visual recording: Did the research use audio or visual recording to collect the data? It is recommended to specify whether the recording of the interview and focus groups has been made by audio or video
- Field notes: Were field notes made during and/or after the interview or focus group?
- Duration: What was the duration of the interviews or focus group? The researchers indicate "researchers executed the discussions and interview, and it took about two hours." It is not clear whether this time refers to the AND group focus interview or only the OR group focus interview. It is recommended to indicate the exact duration of the interview and the average duration of the focus groups separately.
- Transcripts returned: Were transcripts returned to participants for comment and/or correction? It is recommended to justify why this criterion was not performed.
- Software: What software, if applicable, was used to manage the data? It is recommended to explain whether qualitative data analysis software was used or performed manually.
- Comprobación de los participantes: ¿Los participantes proporcionaron comentarios sobre los resultados? Se recomienda justificar por qué no se realizó este criterio.
Además, deben indicar qué miembros del equipo de investigación transcribieron y analizaron los resultados y qué miembros del equipo de investigación revisaron los hallazgos.
Finalmente, con respecto a la recopilación de datos, ¿por qué se realiza una entrevista con uno de los participantes si los principales medios utilizados son los grupos focales? ¿Este participante se negó a participar en los grupos focales? Y si es así, ¿cuál fue el motivo? ¿Crees que esto podría haber influido en los resultados del estudio de alguna manera?
Author Response
Comments and Suggestions for Authors
Hola,
Creo que el artículo debería publicarse si se realizan revisiones menores.
Los autores del estudio afirman haber establecido los criterios de calidad de acuerdo con las recomendaciones de COREQ, pero algunos de ellos no se han cumplido, como la revisión de los participantes y otros no se han explicado claramente. Se requiere una explicación adicional de cada uno de los siguientes criterios COREQ:
Response: We would like to thank you for reading our manuscript and reviewing it. To describe the qualitative research methodology in specific, we have used the COREQ-it is the reporting guideline of focus group discussion (Line 112). We have, however, attempted to clarify the methodology more in detail based on your suggestion.
- Entrevistador / Interviewer / facilitator: What authors conducted the interview or focus group?
Response: We have included a description of the focus group discussion and its facilitator (Line 153~).
- Occupation: What is the occupation and position of researchers?
Response: We have already included a description of the research team in the methodology section (Line 114~). Also, we have included the positions of the researchers (Line 120~).
- Gender: Was the researcher male or female?
Response: We have not described the gender of the research team. We agree on the validity of describing the gender of researchers in certain researches, which has the relevance of the information. We have, however, considered that our research is irrelevant to the gender of the research team.
- Relationship established: Was a relationship established before the start of the study?
Response: We had prepared an environment for rapport building with sufficient ice-breaking time before the focus group discussions and interviews.
- Characteristics of the interviewer: What characteristics were reported about the interviewer / facilitator? For example, biases, assumptions, reasons and interests in the research topic.
Response: We have already described the research team in the methodology section. However, we have included in the methodology section that the research team paid attention to biases and the intervention of it during the data collection process (Line 153~).
- Methodological orientation and Theory: What methodological orientation was stated to underpin the study? e.g. grounded theory, discourse analysis, ethnography, phenomenology, content analysis
Response: We have already described in the analysis of the methodology that the research used content analysis (Line 165~).
- Setting of data collection: Where was the data collected? e.g. home, clinic, workplace
Response: We had rented separate meeting rooms for all the focus group discussions and interviews (Line 150~).
- Description of simple: What are the important characteristics of the sample? e.g. demographic data, date. The researchers only provide information about the sex and age of the participants. It is recommended to provide information on the years of work experience of the participants regarding the phenomenon of study in order to transferability of the data.
Response: We have included the years of work experience of the participants (Table 1) based on your suggestion.
- Interview guide: Were questions, prompts, guides provided by the authors? A table with the semi-structured interview questions is recommended.
Response: As we have described in the methodology section, we had conducted the focus group discussions and interviews using a semi-structured guideline, developed from preliminary studies and more. We have included the translated guideline as a supplement file.
- Audio/visual recording: Did the research use audio or visual recording to collect the data?
Response: We did not conduct an audio/visual recording.
- Field notes: Were field notes made during and/or after the interview or focus group?
Response: We have written a field note in the process of the focus group discussions and interviews. This note was used to clarify the meaning of participants’ statements in the process of analysis (Line 161~).
- Duration: What was the duration of the interviews or focus group? The researchers indicate "researchers executed the discussions and interview, and it took about two hours." It is not clear: interview plus focus groups, or 2 hours each interviews and each focus groups…? It is recommended to indicate the exact duration of the interview and the average duration of the focus groups separately.
Response: The focus group discussions and interviews were conduceted for two hours each.
- Transcripts returned: Were transcripts returned to participants for comment and/or correction? It is recommended to justify why this criterion was not performed.
Response: We were unable to have the participants to review the entire transcripts as the transcripts were about 50 pages in A4 size. Instead, qualitative research experts, occupational and environmental medicine experts, and preventative medicine experts reviewed the transcripts (Line 180~).
- Software: What software, if applicable, was used to manage the data? It is recommended to explain whether qualitative data analysis software was used or performed manually.
Response: We did not use software and analyzed it manually. However, we used the Microsoft Word program for the transcription organization (Line 173~).
- Participant validation: Did the participants provide comments on the results? It is recommended to justify why this criterion was not performed.
Response: We were unable to review the analyzed results by the participants. We have included this as a limitation of the research (Line 398~). Instead, qualitative research experts, occupational and environmental medicine experts, and preventative medicine experts reviewed the transcripts.
- In addition, they should indicate which members of the research team transcribed and analyzed the results and which members of the research team reviewed the findings.
Response: As we described in the methodology section, two research of the research team led the analysis, and the entire team reviewed the results (Line 175~). Separate stenographer transcribed the focus group discussions and interviews, and the facilitator reviewed the content for any missing material (Line 159~).
- Finally, regarding data collection, why is an interview with one of the participants conducted if the main means used are the focus groups? Did this participant refuse to participate in the focus groups? And if so, what was the reason? Do you think this could have influenced the results of the study in any way?
Response: Separate interviews were conducted with participants who had successful experiences in the hearing conservation program. This interview design considered the possible influence of the successful experience of the program to the other participants in the focus group discussion (Line 129~).

Round 2
Reviewer 2 Report
The author answered all the questions from the other reviewers.
Author Response
I would you like to thank you for thoughtful consideration and detailed reviewing of our manuscripts. Through the reviewing, the manuscript was improving and better than before.
Also, we tried to improve the written English language again
We want to say thank you.

Reviewer 3 Report
- The language has greatly improved, allowing for easy and fast reading.
- The text is too extensive, with too many details: it is OK for a brochure but not for a paper:
- Section 1. Socio-demographic characteristics can be easily replaced by a couple of sentences, eliminating also Table 1.
- Section 2. Analysis Result runs for almost three pages! It must be summarized!
- Sections 2.1. The awareness of hearing loss problems and 3.2.2. The overall perception of the hearing conservation program run also for several pages, quoting individual statements. Although of value, those can and must be summarized.
- Same applies to Sections 3.2.3 and 3.3.4: individuals’ statements have only circumstantial value, but should not be in a paper.
- Section Discussion is also too long with many repetitions. Try to halve it in extension!
Author Response
Comments and Suggestions for Authors, and our Responses
1. The language has greatly improved, allowing for easy and fast reading.
Response: We would like to thank you for reading our manuscript and reviewing it. The followings are responses to your comments.
2. The text is too extensive, with too many details: it is OK for a brochure but not for a paper:
Response: We have tried to reduce the overlapping content by reviewing the manuscript as a whole.
• Section 1. Socio-demographic characteristics can be easily replaced by a couple of sentences, eliminating also Table 1.
Response: Instead of eliminating Table 1, we have reduced the sentences describing demographic characteristics to improve readability.
• Section 2. Analysis Result runs for almost three pages! It must be summarized!
Response: As you suggested, we have revised the contents of Table 2 more concisely. The contents deleted from Table 2 are covered in the Result section.
• Sections 2.1. The awareness of hearing loss problems and 3.2.2. The overall perception of the hearing conservation program run also for several pages, quoting individual statements. Although of value, those can and must be summarized.
• Same applies to Sections 3.2.3 and 3.3.4: individuals’ statements have only circumstantial value, but should not be in a paper.
Response: As you suggested, we have also reduced the content of the results more concisely.
• Section Discussion is also too long with many repetitions. Try to halve it in extension!
Response: As you suggested, we have revised Discussion section concisely so that the manuscript do not overlap.
Submission Date
20 February 2020
